# Growing Maize Root: Lectins Involved in Consecutive Stages of Cell Development

**DOI:** 10.3390/plants11141799

**Published:** 2022-07-07

**Authors:** Aliya Aglyamova, Natalia Petrova, Oleg Gorshkov, Liudmila Kozlova, Tatyana Gorshkova

**Affiliations:** 1Kazan Institute of Biochemistry and Biophysics, Federal Research Center Kazan Scientific Center of Russian Academy of Sciences, Lobachevsky Str. 2/31, Kazan 420111, Russia; aliaglyamova@yandex.ru (A.A.); npetrova@inbox.ru (N.P.); gorshkov@kibb.knc.ru (O.G.); kozlova@kibb.knc.ru (L.K.); 2Institute of Fundamental Medicine and Biology, Kazan Federal University, Kremlevskaya Str. 18, Kazan 420008, Russia; 3Institute of Physiology, Federal Research Center Komi Science Center of Ural Branch of Russian Academy of Sciences, Kommunisticheskaya Str. 28, Syktyvkar 167982, Russia

**Keywords:** root growth, plant lectin, transcriptomics, *Zea mays*, *Euonymus europaeus* lectin, calreticulin, jacalin, malectin

## Abstract

Proteins that carry specific carbohydrate-binding lectin domains have a great variety and are ubiquitous across the plant kingdom. In turn, the plant cell wall has a complex carbohydrate composition, which is subjected to constant changes in the course of plant development. In this regard, proteins with lectin domains are of great interest in the context of studying their contribution to the tuning and monitoring of the cell wall during its modifications in the course of plant organ development. We performed a genome-wide screening of lectin motifs in the *Zea mays* genome and analyzed the transcriptomic data from five zones of primary maize root with cells at different development stages. This allowed us to obtain 306 gene sequences encoding putative lectins and to relate their expressions to the stages of root cell development and peculiarities of cell wall metabolism. Among the lectins whose expression was high and differentially regulated in growing maize root were the members of the EUL, dirigent–jacalin, malectin, malectin-like, GNA and Nictaba families, many of which are predicted as cell wall proteins or lectin receptor-like kinases that have direct access to the cell wall. Thus, a set of molecular players was identified with high potential to play important roles in the early stages of root morphogenesis.

## 1. Introduction

Plant morphogenesis is based on individual cell enlargement—a highly dynamic process that involves various aspects such as the increase in cell surface and vacuole. Cell wall extension—the necessary component of cell enlargement—has to be finely tuned in order to coordinate the growth of neighboring cells and to shape the entire organ. Several proteins with lectin domains are reported to be the possible elements for monitoring the cell wall state and evolving signals that regulate organ development [1,2].

Lectin domains are a group of different structural units of proteins capable of specifically recognizing carbohydrate structures [3]. They are found in combination with a wide variety of protein domains and are involved in various physiological processes [4]. Although some lectin motifs are very similar in plants, animals, and bacteria, the combinations of lectin domains with other domains in proteins are most often different in different kingdoms of life and, as a result, the functional role of proteins may vary. In plants, lectins are quite numerous: in total, several hundred genes for proteins with lectin domains have been identified in full-genome studies on higher plant species, namely *Arabidopsis thaliana*, rice, cucumber, soybean, mulberry and flax [5,6,7,8,9,10].

Plant lectins are quite divergent and can be localized in various cell compartments, suggesting differences in their function. The earliest described and best-known lectins, such as Concanavalin A from jack bean (*Canavalia ensiformis*) or phytohemagglutinin from the common bean (*Phaseolus vulgaris*), are those accumulating in large quantities in vacuoles, especially in the storage tissues (cotyledons), to the level of 1% to 8% of the total protein [11]. Another group of lectins has a nucleocytoplasmic location; they are synthesized on the free ribosomes in the cytoplasm, do not possess a signal peptide, remain in the cytoplasm or can be translocated into the nucleus [12]. Such lectins have low abundance in normal conditions of plant development but are induced in the course of stress reactions [13]. A large proportion of lectins are transmembrane proteins; their lectin domains are oriented outside of the plasma membrane, and together with lectins fully located in the apoplast, they have direct access to the cell wall polysaccharides. Such lectins are present in small amounts and can be considered as components of signaling pathways, especially as many of them have kinase domains and are named lectin receptor-like kinases (LecRLK). Some members of the latter group are involved in plant–pathogen interactions [14]. However, proteins with lectin domains located within the cell wall are assumed to participate in the process of cell growth as demonstrated in *A. thaliana* for lectin receptor-like kinases Hercules receptor kinase 1 (HERK1), THESEUS1 (THE1), and FERONIA (FER) [1]. The plant cell wall has a remarkably varied carbohydrate composition, providing a variety of possible ligands that might be the cause of the emergence of a wide variety of plant lectins themselves. The involvement of proteins in lectin domains in the process of cell elongation has not yet received adequate and systemic attention.

The renowned model to study plant growth is the primary root, which is characterized by the presence of consecutive zones that differ in cell sizes and relative rates of their elongation [15]. In proteomic studies, numerous lectins have been identified in roots [16,17]; however, information on their presence can only be found in the Appendix A of the mentioned papers and was not analyzed and discussed. Correspondingly, the dynamics of proteins with lectin domains has never been related to the process of root elongation, especially as the sampling for proteomic studies followed different logics.

Maize (*Zea mays*) root tip contains cell zones at different stages of cell development that can be easily distinguished. Previously, transcriptomic data for five zones of maize root apex were published, including a comprehensive analysis of expressed genes for glycosyltransferases and glycoside hydrolases involved in cell wall metabolism [18,19]. The samples that were also extensively analyzed by immunocytochemistry for cell-wall-related epitopes included root cap, meristem zone, early elongation, active elongation and late elongation zones, making it possible to relate gene expression to specific early stages of root morphogenesis. In the current article, we present the results of genome screening for maize genes that encode proteins with lectin domains and relate their expression to the stage of root cell development and peculiarities of cell wall metabolism. For the first time, we revealed the genes for proteins with lectin domains that are specifically up-regulated at the initiation of cell elongation and are important in the course of active elongation or late elongation, opening the door for further studies on the molecular mechanisms of lectin involvement in plant morphogenesis.

## 2. Results

### 2.1. Genes for Proteins with Lectin Domains in the Maize Genome

We analyzed the presence of proteins with domains characteristic of various lectin families in the maize genome. In addition to the conventional plant lectin families—amaranthins, calreticulins, C-type lectins, the *Euonymus europaeus* lectin (EUL) family, the *Galanthus nivalis* agglutinin (GNA) family, the hevein family, the jacalin-related lectin family, the legume lectin family, the lysin motif (LysM) family, the *Nicotiana tabacum* agglutinin (Nictaba) family and RicinB lectin family [13]—our list was expanded by several other protein families. As was the case for Jiang et al., we added galactose-binding- and galectin-like-domain-containing proteins to the list, since in many non-plant organisms, these domains fully fit the definition of lectins as proteins carrying at least one carbohydrate-binding domain with no catalytic activity toward their own ligands [6,14]. In plant proteins, these domains occur in combination with additional domains characteristic of β-galactosidases and galactosyltransferases, respectively. We also included the malectin and malectin-like families. The malectin domain is classified as CBM (CBM57) in the CAZy database [20], indicating it as a part of a carbohydrate-active enzyme; however, in plants, as distinct from some non-plant organisms, it is not combined with the domains that are enzymatically active on carbohydrates [10,21].

A search for lectin domains in the maize genome yielded 306 putative genes for proteins with lectin domains (Table 1). The identified maize genes encoded members of 15 different families of proteins with lectin domains. No genes were found for the *Agaricus bisporus* agglutinin (ABA) and cyanovirin-N (CV-N) families. These lectins were previously detected among plants only in mosses or algae, respectively [22]. The search for the class V chitinases domain that is characteristic of the CRA family in the maize genome yielded no results.

Four lectin families were the most abundant in the maize genome: GNA, legume, malectin-like and LysM (Table 1). They include numerous LecRLKs, which together with the corresponding lectin domain possess a kinase domain (PF00069) and a transmembrane domain. LecRLKs are also common in the malectin family, whose members are structurally similar to malectin-like proteins but less numerous. Another LecRLK family is C-type lectins, which are common in animals and almost absent in plants [13]. This is the smallest maize lectin family, with only one representative identified (*Zm00001d040192*). A kinase domain was also found in three members of the jacalin-related lectin family (Table 1), although they are probably not receptor-like kinases because they lack a signal peptide and a transmembrane domain.

Among 306 maize genes encoding proteins with lectin domains, 253 were predicted to be localized in the cell wall (termed extracellular in the prediction services) or in the plasma membrane, where they have direct access to cell wall components (Appendix A). Proteins with predicted localization in the plasma membrane were members of the GNA, legume, LysM, malectin-like, malectin, and C-type families, whereas proteins with potential localization in the cell wall belonged to the EUL, galactose-binding, hevein, RicinB, and jacalin-related lectin families. Most of the proteins with direct access to cell wall polymers were represented by transmembrane LecRLKs; they have a lectin domain outside the plasma membrane, whereas a kinase domain faces the cytoplasm.

### 2.2. Expression of Genes of Proteins with Lectin Domains in Various Zones of Growing Maize Root

Over 70% of the genes of proteins with lectin domains (223 of 306) were expressed in growing maize root (TGR over 16 in at least one of the samples) (Table 1). Expression values for all 306 maize genes are presented in the Appendix A. A hierarchical cluster analysis was performed to better understand the expression patterns of the lectin genes in the samples studied. As a result, 223 expressed genes formed five clusters (Figure 1).

The first cluster (Figure 1) included the genes with the highest mRNA levels at the very tip of the primary root, in the root cap and meristem. The genes with the highest TGR values in this cluster were the genes of the jacalin-related lectin family (*Zm00001d019312*) and two EUL genes (*Zm00001d022421* and *Zm00001d028998*) (Table 2). An interesting malectin gene (*Zm00001d027337*) also belonged to this cluster—its ortholog (*At2g22610*) from *A. thaliana* encodes the protein, which in addition to the lectin domain, contains a kinesin motor domain (Table 2). The most significant fold drop in mRNA abundance in the following zone of early elongation was detected for one of the LysM genes (*Zm00001d021447*) (Table 2).

The second cluster consisted of genes with significantly increased mRNA levels in the early elongation zone of maize root (Figure 1). Cluster II included genes encoding calreticulins (*Zm00001d019283*, *Zm00001d003857*, *Zm00001d025305,* and *Zm00001d005460*), which were extremely highly expressed in most of the zones examined but peaked at elongation initiation (Table 2). Some genes encoding β-galactosidases (*Zm00001d022621*, *Zm00001d044290,* and *Zm00001d048440*) and glycosyltransferases (*Zm00001d013052*, *Zm00001d037233*, *Zm00001d041504,* and *Zm00001d049617*) with lectin domains were also in this cluster (Table 2 and Appendix A). The largest fold in expression at the beginning of cell elongation was observed for the genes of the malectin-like (*Zm00001d046838*) and legume (*Zm00001d049741*) families (Table 2).

The genes that were up-regulated during the transition from cell division to cell elongation belonged to several clusters (Figure 1). To identify the genes, which more than doubled their mRNA abundance in the early elongation zone compared to the meristem zone and had a significant level of expression, we used the following criteria: Log2(eElong/Mer) > 1 and TGR (eElong) > 100 (Appendix A). Among 38 lectin genes fitting these criteria, 26 (68%) were protein kinases from various families: 12 from GNA, 6 belonged to malectin or malectin-like, 3 from the legume family, and 2 were from LysM family. Moreover, all three maize jacalins with a protein kinase domain (*Zm00001d010169*, *Zm00001d012208*, and *Zm00001d048876*) fell into this group. Lectins without a kinase domain up-regulated at the beginning of elongation included three genes encoding proteins with LysM domains, three with malectin or malectin-like domains, and two with Nictaba domains; *Zm00001d040190* for the protein containing the EUL domain was most highly expressed in this group (Appendix A). Virtually all of the genes activated in the early elongation zone were additionally up-regulated in the active elongation zone. The number of genes meeting the Log2(Elong/Mer) > 1 and TGR (Elong) > 100 criteria reached 72 (Appendix A).

The third gene cluster was characterized by a high expression during early and active elongation and some decrease in mRNA abundance in the late elongation zone (Figure 1). Similar mRNA level dynamics in growing maize root was observed for the genes of cellulose synthases responsible for primary cell wall formation [18]. This cluster contained some LecRLK genes from the malectin-like family (e.g., *Zm00001d029047*, *Zm00001d047533*, *Zm00001d018789* homologous to FER and HERK), Nictaba family genes (e.g., *Zm00001d029673*) and jacalin-related lectins (e.g., *Zm00001d010169*). The only EUL gene (*Zm00001d040190*) belonging to this cluster was highly expressed (Table 2). Some of the above mentioned genes (FER *Zm00001d029047*, Nictaba *Zm00001d029673*, EUL *Zm00001d040190*) and several others (e.g., GNA *Zm00001d043252*) were co-expressed with the genes of cellulose synthases associated with the primary cell wall (Appendix A).

The fourth cluster was characterized by the expression opposite to that of cluster I (Figure 1). It contained the largest number of expressed genes: 59. For these, mRNA levels increased from the root tip to the late elongation zone. This cluster contained two other members of the calreticulin family (*Zm00001d038648* and *Zm00001d012170*); however, these had a much weaker expression than the members of this family in clusters I and II. Some genes encoding members of the malectin (e.g., *Zm00001d002313* and *Zm00001d026306*) and malectin-like (e.g., *Zm00001d027439*) families also showed a similar expression pattern (Table 2).

The fifth cluster included genes whose expression was the highest in the zone of late elongation of maize root (Table 1). Although the fifth cluster included quite a few genes, their expression was relatively low. The highest value was shown by β-galactosidase with lectin domain (*Zm00001d042654*) (Table 2 and Appendix A). Genes that were co-expressed with the genes of secondary cell wall cellulose synthases belonged to the fourth and fifth clusters. These included members of calreticulins (maize homolog of CRT3, *Zm00001d038648*), GNA (*Zm00001d007789*), LysM (*Zm00001d027533*) and some other families (Appendix A).

### 2.3. Phylogenetic Analysis of Gene Families Encoding Proteins with Lectin Domains

We performed a phylogenetic analysis to further characterize maize families of proteins with lectin domains and to identify their members homologous to the described representatives of corresponding families from other plant species. The major focus was given to the families with a high proportion of the genes that were actively and differentially expressed in the analyzed samples. As such, the remarkable one is the plant-specific *Euonymus europaeus* lectin (EUL) family that contains eight genes in maize. All identified members were expressed in growing maize root, and six of them belonged to the top 10 groups from various clusters (Table 2). In maize, EUL domains (in protein databases the EUL domain, PF14200, is still designated as Ricin_B-like_lectin, which should not be confused with the lectin domain of RicinB family, PF00652) are not combined with any other domains within one protein (Appendix A), similar to other plant species [1,22]. Double EUL domains were detected in three maize EULs (Appendix A). In dicots, EULs are represented by 1–2 genes carrying one lectin domain; the monocot EUL family is more complex [23]. In the phylogenetic tree, genes of monocots and dicots were organized into separate clades, and the same was true for the genes containing single (EULS) and double (EULD) domains (Figure 2).

An exceptionally high expression level was detected for the calreticulins (Table 2), which are evolutionarily conserved lectins mainly localized in the endoplasmic reticulum. Maize calreticulins are divided into several groups on the phylogenetic dendrogram (Figure 3). Calreticulins 1/2, also known as CRT1a/CRT1b, form one group, while calreticulins 3 form two other groups. Additionally, there is a group of calnexins—membrane-bound versions of calreticulins. Calreticulins 1 and 2 are paralogs of each other and correspond to the two types of animal calreticulins, while type 3 calreticulins are specific for plants [24]. Although calreticulins are considered to be constitutively expressed genes, only about half of all maize genes were expressed in the primary root (Appendix A). Five calreti-culin genes with no expression in maize roots had a length of amino acid residues that was much shorter than the estimated average length of plant calreticulins (420 aa) [25], so their functional integrity can be questioned. Interestingly, maize calnexins (*Zm00001d003857* and *Zm00001d025305*) and calreticulins 1/2 (*Zm00001d019283* and *Zm00001d005460*) had a similar expression pattern, while two expressed type 3 calreticulins (*Zm00001d012170* and *Zm00001d038648*) had a completely different expression pattern and were considerably activated at the onset of cell elongation (Figure 3, Appendix A).

A pronounced differential expression was demonstrated by many members of structurally related malectin and malectin-like families (Table 2). The malectin-like domain consists of two tandem malectin domains [26]. Malectins are widely distributed in nature (in particular, in animals), while the malectin-like domain is found exclusively in plants (*Archaeplastida*) [22]. In the phylogenetic tree, proteins containing malectin or malectin-like domains were separated into different groups (Figure 4), similar to *A. thaliana* genes [26]. Malectin-like proteins themselves were subdivided into two groups according to domain organization. The first group, usually called CrRLKs (*Catharanthus roseus* RLK1-like kinases), included only members with MLD (malectin-like domain) and protein kinase (PK) domain. Another group included mostly LRR-containing proteins with different domain organization. In the extended phylogenetic tree for malectin-like proteins from several plant species, the distinct clades containing only monocots’ or dicots’ genes were also quite pronounced (Appendix A).

In maize, 9 malectins and 32 malectin-like proteins were identified (Table 1). They are mostly receptor-like kinases that sometimes carry leucine-rich repeats involved in protein–protein interactions (Appendix A). All but one member were predicted to be localized in the plasma membrane. The malectin domain of this exceptional member (*Zm00001d027337*) is attached to kinesin, and in the phylogenetic tree, it was allocated to a separate clade together with two genes of *A. thaliana*, *MDKIN1* (*At1g72250*) and *MDKIN2* (*At2g22610*) (Figure 4). Maize MDKIN showed more similarity to MDKIN2 of *A. thaliana* that is angiosperm-specific, while MDKIN1 was present in a wider range of plants [26]. One more small, separated group in the phylogenetic tree (Figure 4) was formed by two maize genes (*Zm00001d039911* and *Zm00001d021566*) and their *A. thaliana* homologs (*At1g28340* and *At1g25570*). Both maize genes belonged to the second cluster, shared the same MLD LRR organization and lacked the kinase domain (Appendix A).

Three maize genes (*Zm00001d029047*, *Zm00001d047533*, and *Zm00001d002175*) showed a high homology to *FERONIA* (FER, *At3g51550*). FER is well-known for its involvement in cell growth and elongation, as are some other malectin-like kinases, like HERCULES 1/2 (HERK1/2, *At3g46290*/*At1g30570*) or THESEUS (THE1, *At5g54380*) (Shinya et al., 2012). Three maize homologs (*Zm00001d037544*, *Zm00001d046838* and *Zm00001d028686*) of *HERK1* (*At3g46290*) and two homologs (*Zm00001d002282* and *Zm00001d018789*) of *HERK2* (*At1g30570*) genes were also highly expressed in the studied samples. However, no close homolog of *THE1* (*At5g54380*) was found in the maize genome (Figure 4), and the same was true for rice [27].

Jacalin-related lectins, which are mostly mannose-specific proteins with a localization in the cytoplasm or nucleus [13], have a monocot-specific combination of the lectin domain with dirigent or kinase domain [4,13]. In maize, a combination of lectin domain with the dirigent domain (PF03018) or kinase domain (PF00069) was also detected (Appendix A). The number of genes in this family varies greatly from species to species. In maize, 20 genes of the jacalin-related lectin family were found (Appendix A), while 50 genes were described in *A. thaliana*, 4 in flax, 8 in cucumber and 28 in rice [5,6,7,10]. Jacalins of dicots and monocots were mostly well-separated in the phylogenetic tree (Figure 5). Two maize genes of dirigent jacalin-related lectins (*Zm00001d005472* and *Zm00001d019312*) showed a close homology to the wheat gene *TaJA1* (*TaJRL4*) (Figure 5). One of them (*Zm00001d019312*) encodes β-glucosidase-aggregating factor (BGAF); it showed an extremely high expression in root cap and meristem (Appendix A). Dirigent proteins are ubiquitous in plants and are known for their role in the regio- and stereoselective coupling of phenoxy radicals (e.g., during lignan biosynthesis) [28]. However, dirigent-like TaJA1 is functionally distinct from wheat dirigent protein TaDIR13 and does not have any effect on lignan content in plants [29].

Some previously identified jacalin-related proteins were experimentally determined to have intracellular localization [30]. However, most members of the maize jacalin-related lectin family were predicted to be extracellular (Appendix A). Apoplast-located jacalin-related proteins were experimentally identified earlier in several other species and described as unconventionally secreted lectins [31,32]. Though several genes of the jacalin-related lectin family encode proteins carrying the protein kinase domain, they do not appear to have a transmembrane domain (Appendix A); therefore, they are considered soluble proteins.

Dendrograms for the rest of the families, members of which were rather poorly expressed in growing maize root, are presented in Appendix A. Among them, one of the largest families of maize lectins is the legume family, with 51 identified members. Mostly, legume lectin-domain-containing proteins are abundant in seeds and are present in low concentrations in other plant organs [14]. Indeed, more than half of the legume genes were not expressed in the studied zones of maize root, and the other half were had a relatively low expression in comparison to members of other lectin families (Appendix A). A similar situation occurred with the Nictaba proteins; they are known as nucleocytoplasmic stress-inducible lectins and are hardly detectable in plants under unstressed conditions [14]. In maize root, all 18 identified genes were expressed (TGR > 16 in at least one sample), although several had an extremely low level (TGR about 20) (Appendix A). According to phylogenetic analyses, some genes of maize and *A. thaliana* are grouped into distinct clades, both in legume and Nictaba families (Appendix A).

The largest lectin family in maize, GNA family with 81 identified members, included mostly chimeric proteins containing the mannose-specific lectin_B domain, EGF-like domain (epidermal growth factor-like), S-locus glycoprotein domain, PAN/Apple domain and protein kinase domain (Appendix A). The EGF-like domain contributes to disulfide bond formation, the S-locus glycoprotein domain is known to be involved in plant self-incompatibility, and the PAN/Apple domain provides interactions with specific carbohydrates or proteins [4,33,34]. Highly expressed maize GNA genes were in clusters III and IV, showing the highest levels of mRNA in the active or late elongation zones of the primary root (Table 2). According to the phylogenetic analysis, the maize GNA genes were divided into three major clades (Appendix A). The first clade included 24 maize genes with only 2 genes of *A. thaliana*. The second clade (32 maize genes) included genes with different domain organization, mostly lacking the S-locus glycoprotein domain. The third clade (25 maize genes) was subdivided into three subclades: one containing only *A. thaliana* genes and two in which *A. thaliana* and maize genes were mixed. Two of the three genes encoding GNA proteins that had significant mRNA levels were co-expressed with the cellulose synthase genes of the primary and secondary cell wall (Appendix A), which may indicate their possible involvement in the synthesis or maintenance of this compartment.

As distinct from members of the described above families, clades of the LysM family contained both genes of dicots and monocots (Appendix A). This family mainly contains lectin receptor-like kinases or proteins with a role in the defense against pathogens. They contain one or a few tandem lectin domains that have a carbohydrate-binding activity toward chitin, chitin oligomers or peptidoglycan [14]. Among maize genes, one gene was found to have an F-box domain that is mostly involved in protein ubiquitination (Appendix A) [35]. In maize roots, LysM genes were mostly expressed in the late cell elongation zones, with a few exceptions. *Zm00001d021447,* which was identified as an ortholog for *LYM2* (*LysM domain-containing GPI-anchored protein 2*, *At2g17120*) gene of *A. thaliana,* showed a high expression in the root cap zone (Table 2). Another maize homolog of *LYM2*—*Zm00001d027533* was actively expressed in the elongation and late elongation zones. Several homologs of the experimentally characterized receptor-like kinases of *A. thaliana* that belong to the LysM family were also identified in maize (Appendix A). Among them was *Zm00001d031676*—the homolog of *CERK1* (*At3g21630*) that is a part of the plant’s innate immune system [14].

## 3. Discussion

### 3.1. Numerous Maize Lectins Have Access to Cell Wall Polysaccharides

Generally, the genome of a higher plant contains a large number of genes for proteins with lectin domains distributed over numerous different families. The screening of the maize genome revealed over 300 genes carrying lectin domains; half of these genes encode LecRLKs (Table 1). All maize members of GNA, legume, malectin-like, C-type families and most of LysM and malectin proteins are predicted to be membrane-bound with a lectin domain facing the extracellular space. Therefore, they are potentially able to directly interact with cell wall polysaccharides. In addition to transmembrane proteins, whose lectin domains have access to cell wall carbohydrates, there are members of the jacalin-related lectin, galactose-binding, hevein, RicinB and EUL families that are predicted as cell wall proteins (Appendix A).

The cell wall is the main carbohydrate-containing structure of plants, and it is logical to assume that lectin domains, as carbohydrate-binding elements, interact with cell wall polysaccharides. The hierarchical clustering of data on the expression level of all the genes encoding proteins, which have lectin domains in various tissues of flax plants, grouped the analyzed genes according to the cell-wall-type characteristic of the studied samples [10]. This indicates the possibility of interactions between cell wall polymers and proteins with lectin domains. In maize, the co-expression of several genes for lectin proteins from various families with the genes for cellulose synthases of the primary and secondary cell wall was well-pronounced.

However, the direct data on such interactions is scarce. Using an in-gel affinity electrophoresis assay, electrophoresis shift assay, dot-blot analysis and ELISA, several proteins from the malectin-like family and their lectin domains were shown to interact with homogalacturonan; those included FER, as well as its individual malectin domains, ANX1/2, and BUPS1 ectodomains [36]. Further studies reported that the extracellular domain of FER (particularly MALA) specifically binds to de-methylesterified pectin both in vitro and in vivo [37]. In contrast to the above works, the results of the crystallographic resolution of ANX1/2 structures and their comparison with other CBMs suggest that these proteins lack the conserved binding surfaces for carbohydrates and can be designed for other ligands [38,39]. Attempts to assess the binding of these proteins to various carbohydrates (including low methyl-esterified pectin) using isothermal titration calorimetry revealed no such interactions [38]. Thus, the question of the carbohydrate ligands of CrRLK1s, as well as those of other proteins with lectin domains, in plant cell walls remains unanswered.

The effective approach to characterize the presence and specificity of a protein binding to various carbohydrates is the application of glycan microarrays [40,41]. Such arrays contain hundreds of spots with individual glycans for the simultaneous analysis of their interactions with the protein under study. The developed glycan microarrays are mainly focused on the importance of protein glycosylation and mostly contain the oligosaccharidic chains of N-glycosylated proteins. The accumulated data in this field already enable machine-learning to be performed for the annotation of lectin specificity toward various glycans present in N- and O-glycosylated proteins [42]. However, such microarrays barely contain any fragments of plant cell wall polysaccharides and, up to now, have not been used to check the ability of plant lectins to bind cell wall glycans.

In addition, the provided predictions based on the approaches of bioinformatics often differ in various services. For example, most maize jacalin-related proteins are predicted as extracellular by the LocTree3 service; however, they are mostly regarded as cytoplasmic by the DeepLoc service (Appendix A). Members of the EUL family are also predicted to be localized in the cytoplasm or cell wall according to the two distinct prediction services (Appendix A).

Altogether, plant proteins carrying lectin domains have an obvious potential to interact with cell wall carbohydrate molecules. However, our knowledge on such interactions is currently rather poor, and there is a high demand for research in this direction.

### 3.2. Comparison of Lectin Sets in Monocots and Dicots

The genome-wide identification of genes encoding proteins with lectin domains was performed for several dicotyledonous plants: cucumber, soybean, mulberry, *A. thaliana*, and flax; and two monocotyledonous species: rice and maize (current paper), allowing for some generalizations [5,6,7,8,9,10]. Evidently, the number of proteins with a lectin domain in various plant families is variable and species-specific, but it is not dictated by plant membership in one class or another. The only noticeable exception is the EUL family, which is represented by 1–3 genes in dicotyledons and by 5–10 genes in monocotyledons [43].

Nevertheless, significant differences in the domain organization of proteins with lectin domains in dicotyledons and monocotyledons are observed. In particular, the combination of the jacalin domain with the dirigent and kinase domains is the specificity of monocot proteins [44]. Within the EUL family protein, the EUL domain is not combined with any other domains, but it may be single (designated as EULS) or double (designated as EULD) [43]. Tandem EUL domains are found only in monocotyledonous plants and form a separate clade on the phylogenetic tree (Figure 2). Moreover, maize and rice EULS are also separated from the EULS of dicots into a distinct group (Figure 2).

In addition, in the phylogenetic trees of many families, there are several clades containing only monocot genes or only dicot genes, although they may have similar domain organization, as was shown for the malectin-like, legume, GNA and Nictaba families (Appendix A). A feature of the CrRLK set of genes in cereals is that they have no orthologs of the *THESEUS1* gene in their genomes (Appendix A) [27]. THE1 is required for hypocotyl cell elongation, responses to cell wall damage induced by the inhibition of cellulose biosynthesis, and during plant–pathogen interactions [45]. Perhaps, in cereals, some other protein fulfills functions of THE1.

Some of the differences mentioned, at least in those proteins whose lectin domains have access to cell wall polysaccharides, may be related to peculiarities of cell wall composition. Maize, like rice, has primary cell walls of type II enriched with mixed-linkage glucans and arabinoxylans, at least at the stage of cell elongation, in contrast to dicotyledonous species whose cell walls are poor in these components [46].

### 3.3. Lectin Genes with Zone-Specific Up-Regulation in Growing Maize Root

In total, 70% of all identified genes for proteins with lectin domains were expressed in the analyzed zones of maize root. In most cases, genes from the same family have different expression patterns. Each zone has its own set of specifically up-regulated genes of proteins with lectin domains, both related and unrelated to the cell wall.

#### 3.3.1. Meristematic Zone

The zone with active cell divisions is characterized by the up-regulated transcription of several EUL genes, namely *Zm00001d028998*, *Zm00001d022421*, and *Zm00001d024982* (Table 2). The most up-regulated in maize meristematic zone genes for proteins with EUL domains are the homologs of *OsEULD1a* (*Zm00001d028998*), and *OsEULD1b* (*Zm00001d022421*) (Table 2). In rice, *OsEULD1a* and *OsEULD1b* are also highly expressed in the tips of roots, both primary and lateral [23]. A glycoarray analysis revealed that the EUL domain of the sole *A. thaliana* lectin of this family, ArathEULS3, interacts with N-glycan structures containing Lewis Y, Lewis X and lactosamine; its specificity differs from that of *Euonymus europaeus* lectin, despite the high sequence identity of these domains [47]. The rice lectin OsEULD1a preferentially binds to galactose-related sugars [48]. Though the EUL family members are often considered as the nucleocytoplasmic proteins involved in plant defense, in *A. thaliana*, the only member of the EUL family is involved in ABA-induced stomata closure in leaves [14,49]. It could also be important for the inhibition of cell division or cell elongation controlled by ABA signaling in the root cap under osmotic stress conditions [50]. In rice, all five OsEULs are located in the nucleus, although they lack a classical nuclear localization site, while two are also present in the cytoplasm [23]. In summary, the maize root meristem expresses two genes encoding EUL proteins whose specificity, function and localization are not clear yet and require further studies.

Active expression in the zone of cell division was also detected for the maize gene (*Zm00001d027337*) encoding a kinesin carrying malectin domain protein. This unwanted combination of two distinct domains is specific to plants and can be found in dicots, monocots, and early vascular plants [51]. Two malectin domain kinesins (At1g72250, At2g22610) were reported in *A. thaliana* and one was found in the genome of *Z. mays* (Appendix A). *Malectin domain kinesin 2* (*MDKIN2*, *At2g22610*) and its close homologs are actively expressed in the zones of cell proliferation in different organs in *A. thaliana* and flax [10,51]. Between cell divisions, it is predominantly localized in the nucleus, while during cell division, it appears close to the tubulin spindle or at the phragmoplast midzone [51]. The co-expression of genes for malectin with a kinesin domain and those encoding several cell-cycle-dependent proteins were revealed in flax [52]. *MDKIN2* knockdown mutants showed abnormalities in the development of ~60% of seeds with pleiotropic and stochastic defects. However, no evident effect on root growth was reported [51]. Although it is now believed that MDKIN2 protein is important but non-essential in organs, except for seeding and producing pollen, it is still possible that its detailed study may reveal some unknown features. One more gene encoding the malectin-domain-containing protein was also up-regulated in the maize root meristem (Table 2).

The gene for the jacalin-related protein BGAF1 (β-glucosidase-aggregating factor, *Zm00001d019312*) was found to be actively expressed in the tip of maize root (Table 2). It encodes dirigent domain-containing jacalin-related lectin that specifically interacts with maize β-glucosidase isozymes ZmBGLU1 (*Zm00001d023994*) and ZmBGLU2 (*Zm00001d024037*), forming large insoluble complexes [53]. ZmBGLU1 was characterized by exceptionally high expression in growing maize root; however, in contrast to BGAF1, the highest levels of ZmBGLU1 mRNA were observed in the active and late elongation zones [19]. BGAF1 was identified as a polyspecific lectin with a marked preference for Gal. The physiological role of both BGAF1 and its complex with β-glucosidases is not clear [53]. Interestingly, the ability to form such protein–protein interactions may be unique to maize BGAF1 since the homolog protein from sorghum did not show any specificity toward glucosidases [54]. The close homolog of BGAF1 is wheat jacalin-related lectin TaJA1 (TaJRL4), which is known to confer a basal but broad-spectrum resistance to pathogens [55]. Another homolog is the rice lectin OsJAC1 (LOC_Os12g14440). The expression of its gene was found in young roots and coleoptiles, sheaths, leaves, and nodes of rice stems [56]; injury or pathogen attack led to its significant activation [57]. The overexpression of OsJAC1 resulted, on the one hand, in the suppressed elongation of cells in coleoptiles and internodes [56]. However, on the other hand, the overexpression of OsJAC1 in rice led to a quantitative broad-spectrum resistance against different pathogens, including bacteria, oomycetes and fungi [57]. Afterwards, the possible role of OsJAC1 in resistance to gamma radiation was revealed, and OsJAC1 was suggested to be a key player in DNA damage response in plants [58].

Altogether, the genes up-regulated in the meristematic zone mainly encode intracellularly localized proteins. The proportion of lectin receptor kinases in cluster I, at least those with TGR > 100, was significantly lower than the average for all lectins. In the meri-stematic zone, they constituted 32% (7 of 22) compared to 51% across the whole genome (156 of 306). In the top 10 genes from cluster I (Table 2), only *Zm00001d026303* of the malectin family encodes a protein sequence that has the domain structure of a lectin receptor-like kinase (Appendix A).

#### 3.3.2. The Early Elongation and Active Elongation

The onset of the cell elongation process in maize primary root is coupled to pronounced changes in the expression profiles of several genes for proteins with lectin domains. The highest expression among all genes for such proteins was detected for calreticulins (Table 2), which are known as highly conserved and ubiquitously expressed proteins located in the endoplasmic reticulum and acting in glycoprotein folding quality control and Ca^2+^ homeostasis [25,59]. The phylogenetic dendrogram divides maize calreticulins into several groups (Figure 3). Calreticulins 1/2, also known as CRT1a/CRT1b, form one group, while calreticulins 3 form two others. Calnexins, membrane-bound versions of calreticulins, join the fourth group. Two distinct groups of calreticulins (CRT1/2 versus CRT3) were also found by phylogenetic analyses and expression studies in *A. thaliana*, rice, wheat, Polish canola, and pine (*Pinus taeda*) [25,60]. The set of genes activated at the beginning of elongation of maize root cells included two calnexins (*Zm00001d025305* and *Zm00001d003857*) and two calreticulins 1/2 (*Zm00001d019283* and *Zm00001d005460*) (Table 2). Calreticulins 1 and 2 of *A. thaliana* were previously reported to be actively expressed in root tips and other expanding parts of plants [61]. The abundance of calreticulins 1/2 in young tissues can be connected to increased protein synthesis during active growth. Such proteins may include those necessary for cytoskeleton and cell wall reorganization, aquaporins and other transporters, transcription factors, hormone-related proteins, etc. [17].

Among the 38 genes that are more than twofold up-regulated at the transition from cell division to cell elongation (Log2(eElong/Mer) > 1, TGR (eElong) > 100) (Appendix A), the predominant group was represented by protein kinases (68%). Out of them, nine genes contained malectin or malectin-like domains. They belong to different clades on the phylogenetic tree (Figure 2); all of them further increase their expression during the active elongation stage and are predicted to be localized in the plasma membrane (Appendix A).

The homolog of the *RGIR1* (*root growth inhibition receptor 1*) gene of *A. thaliana*, *Zm00001d052306*, showed a high expression in all zones of the maize root with an up-regulation in the early elongation zone (Table 2). *RGIR1* is known as a positive regulator of root growth that preferentially acts in the root elongation zone [62]. The maize homolog of RGIR1 contains a malectin domain together with kinase, transmembrane and leucine-rich repeat domains (Appendix A), similar to the protein of *A. thaliana* that was isolated from plasma membrane. Knockdown mutants *rgir1-1* have shortened main roots; this phenotype is associated with a lower elongation rate and decreased cortex cell number in the transition zone and elongation zone of the root tip [62]. The maize homolog of *RGIR1* may also be involved in root system architecture maintenance.

Two maize homologs of *HERK2* and two of *FER* were significantly up-regulated in the course of active elongation and belonged to cluster III (Table 2 and Appendix A). These receptor-like proteins show amazing versatility in terms of control of plant development. FER is known for its role in the tip growth of root hairs and trichomes, cell expansion in roots and petioles, definition of cell shape in the leaf epidermis, maternal control of fertilization and responses to abscisic acid (ABA), among other functions [1,36,63,64,65]. *FER* is expressed in almost all tissues of *A. thaliana*, but its expression is increased in the zones of cell expansion. The same observations were made for *HERK1* and *2* [1]. The small secreted peptide rapid alkalization factor (RALF) was identified as its protein ligand [65]. Presumably, FER, HERK, and their related proteins function as mechanosensors, regulating cell wall extensibility in multiple ways [66]. The maize homologs of *FER* and *HERK2* gradually increased their expression toward the zone of active elongation and decreased in the zone of late elongation (Table 2 and Appendix A). The same expression pattern was demonstrated by the primary cell wall cellulose synthase genes [18]. One homolog of *FER* was even co-expressed with these cellulose synthases (Appendix A). An increase in the area of the primary cell wall requires not only new portions of structural polysaccharides but also the provision of an increasingly larger surface area with the necessary sensors.

Maize has three homologs of *FER*; one of them, *Zm00001d002175*, had a different expression character and in phylogeny was grouped with five rice genes (*LOC*_*Os05g25430*, *LOC_Os04g49690*, *LOC_Os05g25450*, *LOC_Os05g25350*, *LOC_Os05g25370*), while the other two maize genes (*Zm00001d047533* and *Zm00001d029047*) were grouped with two remaining rice genes (*LOC_Os01g56330* and *LOC_Os03g21540*) (Appendix A). It is not clear whether the proteins encoded by the genes from this separate subclade have the same function as FER, or whether they have features unique to monocots.

*Zm00001d040190* (homolog of rice *OsEULS3*) from the EUL family was considerably up-regulated and had a high expression level (Table 2 and Appendix A). It was also co-expressed with primary cell wall cellulose synthase genes. This correspondence between the expression of the EUL lectin gene and the cellulose synthase genes awaits an explanation.

Another family of genes whose differential expression in the growing root of maize was unexpected is GNA. The GNA family in maize is huge, but its members mostly do not have significant TGR values in the growing root. Nevertheless, three genes were in the top 10 of clusters III and IV (Table 2). Two of these genes (*Zm00001d021729* and *Zm00001d043252*) encode LecRLKs containing five domains (Appendix A). The highest mRNA content of both of these genes was found in the zone of active elongation growth, with a further decrease in the zone of late elongation. The *A. thaliana* homolog of *Zm00001d021729* (*At4g21390*) encodes LecRLK B120. Its function remains unclear; however, it shows a 200-fold up-regulation in response to apoplastic reactive oxygen species [67]. The latter are well-known mediators of cell wall loosening and stiffening during maize root growth [68]. The second of the genes for GNA family proteins, which was up-regulated in the zone of active elongation, was co-expressed with cellulose synthases of the primary cell wall.

#### 3.3.3. The Late Elongation

Many maize genes encoding proteins with lectin domains were up-regulated in the active elongation zone and further increased their mRNA levels in the late elongation zone. These genes were assigned to clusters IV or V (Table 1). In the zone of late elongation, cells of maize root gradually slowed down their growth due to strengthening of the cell wall, cells of vascular tissues deposit a secondary cell wall, and root hairs appear soon after elongation is terminated [18,69].

Maize genes belonging to cluster V and/or co-expressing with the secondary cell wall cellulose synthases have the highest level of mRNA abundance in the zone of late elongation (Table 2 and Appendix A). Representatives of Gal-binding, calreticulins 3, and several GNA encoding genes belonged to this group.

β-Galactosidases (EC 3.2.1.23) mediate the hydrolysis of terminal non-reducing β-D-galactose residues in β-D-galactosides. All plant β-galactosidases belong to the glycoside hydrolase (GH) family 35. Thirteen out of sixteen maize β-galactosidases were predicted to have a galactose-binding lectin domain (Appendix A). Thus, GH35 proteins and proteins containing the galactose-binding lectin domain are roughly the same. The expression of genes encoding GH35 members in the growing maize root was previously described [19]. It is important to note here that at least one of these genes belonged to the top 10 of each expression cluster (Table 2). This means that not only late elongation but any stage of maize root cell development was characterized by a high level of transcripts for one or another β-galactosidase isoform.

Two calreticulins 3 (*Zm00001d012170* and *Zm00001d038648*) were up-regulated in the active and late elongation zones of maize roots. Calreticulins 3 are specific for land plants and differ from the more common calreticulins 1/2 in the structure of the C-domain [25]. This domain provides the interaction of calreticulin 3 with the misfolded form of brassinosteroid receptor BRI1 and its retention in the endoplasmic reticulum [25,70]. BRI1 homologs, BRL1 and BRL3, are involved in vascular tissue differentiation [71]. It is possible that the expression of calreticulins 3 in maize root reflects the development of the phloem and xylem cells. This is also indicated by the co-expression of one of the maize CRT3 with the cellulose synthase genes of the secondary cell wall (Appendix A).

Genes of the extensive but poorly characterized GNA lectin family mostly had a low expression in the growing maize root. However, at least one of them (*Zm00001d007789*) demonstrated a remarkable up-regulation in the late elongation zone. It has no characterized homologs and, in contrast to GNA members that are up-regulated in the active elongation zone, encodes a sequence containing only three domains. This gene was co-expressed with cellulose synthases of the secondary cell wall. This co-expression may indicate that GNA LecRLKs encoded are required for the construction of a particular type of cell wall.

## 4. Materials and Methods

### 4.1. Identification of Genes for Proteins with Lectin Domains in the Maize Genome

The maize genes for proteins containing lectin domains were identified by name search of characteristic pfam domains (Pfam 34.0 database) in the Phytozome v13 database (*Zea mays* genome assembly RefGen_V4) [72].

The pfam domain names that were used to identify genes are listed in Table 1. Since there is no pfam number for the CRA lectin family, members of this family were searched for the presence of cd02879 (GH18_plant_chitinase_class_V) domain using the CD-search tool [73]. Nevertheless, not a single gene was found. Domain searches for PF14200 (EUL) and PF00652 (RicinB) yielded similar results, so an additional search using PTHR ID was performed for the EUL family.

Protein sequences of maize genes were obtained from the Phytozome database. The presence of lectin domains in the proteins encoded by identified genes was confirmed using the web CD-search tool [73]. The domain organization of maize lectins was resolved using the web tool InterProScan [74].

Amino acid sequences encoded by some genes were re-predicted in the web service Augustus in cases of the presence of non-characteristic domains in the sequence [75]. In cases where Augustus did not give a satisfactory result, the amino acid sequence was re-predicted using the FGENESH web service [76]. If, for any reason, a reliable sequence was not obtained, the sequence taken from the Phytozome database was used. In cases where amino acid sequences were found to not carry lectin domain after analysis with the CD-search and InterProScan services, they were excluded from further study.

All protein sequences were checked both for the presence of a signal peptide with the SignalP-5.0 and TargetP services and for the presence of a transmembrane domain in the TMHMM v2.0 service [77,78,79]. The putative subcellular localization of proteins with lectin domains was predicted using the web services LocTree3 and DeepLoc-1.0 [80,81].

### 4.2. Phylogenetic Analysis

In addition to maize genes, the genes of *A. thaliana* and *Oryza sativa* were taken from the Phytozome v13 database for phylogenetic analysis. The search was carried out by the pfam domain name and characteristics of a certain lectin family (Table 1). Protein sequences of wheat, soybean and flax genes were taken from previous studies [6,10,44].

The multiple alignment of protein sequences was performed using the ClustalW web service for further phylogenetic analysis [82]. The sequence alignments were used to build phylogenetic trees in the IQTREE1.6.12 software using maximum-likelihood phylogenetic analysis [83]. The best-fit model was automatically selected by ModelFinder, and the best-fit model of sequence evolution was selected using the Bayesian Information Criterion (BIC) [84]. The ultrafast bootstrap branch support with 10,000 replicates was used to construct each dendrogram [85]. Only trees with a Robinson–Foulds distance equal to or less than 4 were accepted as reliable. In the case of an additional tree (Appendix A), a distance value equal to 6 was considered as appropriate since it included a large number of sequences.

The trees were visualized using the iTOL 6.3.2 web service and edited in Adobe Illustrator CC 2017 [86]. Their homology was defined using built trees and protein BLAST [87]. The *A. thaliana* and *O. sativa* gene descriptions were taken from The Arabidopsis Information Resource (https://www.arabidopsis.org/index.jsp, accessed on 15 April 2022) and the funRiceGenes dataset [88].

### 4.3. Analysis of Transcriptomic Data

The previously obtained transcriptomic data for 5 zones of the apical part of the primary maize (*Zea mays* L.) root (before the root hairs initiation) (Figure 1) were used for the analysis of gene expression [18]. Zones that contained cells at different stages of development were distinguished depending on the distance from the root apex: the root cap, the meristem zone (the zone of cell division), the zone of early elongation, the active elongation zone and the late elongation zone (Figure 1A). Plant material, RNA extraction, sequencing procedures and initial steps of transcriptomic analysis were previously described [18,19]. Raw data as fastq files were uploaded to the Sequence Read Archive under BioProject accession number PRJNA639682. Briefly, after pre-processing with the BBDuk utility (https://jgi.doe.gov/data-and-tools/bbtools/bb-tools-user-guide/bbduk-guide/, accessed on 5 May 2022), the resulting clean reads were mapped to (against) the genome using the StringTie v2.0 software [89]. DESeq2′s statistical model was used for normalization and differential expression analysis by the R package DEseq2 v1.30.1 [90].

In total, 26,936 out of 44,146 genes were considered to be expressed according to the cut-off total gene reads count (TGR) that was more or equal to 16 at least in one zone [91]. For a better understanding of the main expression patterns of genes encoding proteins with lectin domains, hierarchical clustering, along with a heatmap analysis, was carried out using R packages (cluster, ggplot2, heatmap.2 et al.) in RStudio v.1.4 [92].

To correlate the expression of genes of proteins with the lectin domain and the synthesis of primary and secondary cell walls in maize root, co-expression networks with cellulose synthases were constructed using the CoExpNetViz (Comparative Co-Expression Network Construction and Visualization) tool. Pairwise similarity between gene expression profiles was calculated using the Pearson correlation measure (0.995) and a set of “bait” genes. Two groups of bait genes were used: six genes encoding putative cellulose synthases of the primary cell wall (*Zm00001d019317*, *Zm00001d037636*, *Zm00001d005250*, *Zm00001d019149*, *Zm00001d009795,* and *Zm00001d005478*) and four genes encoding putative cellulose synthases of the secondary cell wall (*Zm00001d043477*, *Zm00001d032776*, *Zm00001d020531,* and *Zm00001d005775*) [18]. The resulting co-expression networks were designated as primary and secondary cell wall modules (PCW and SCW), respectively.

## 5. Conclusions

The early stages of maize root morphogenesis employ hundreds of proteins with lectin domains. Such proteins are highly diverse and belong to all families found in higher plants; they can be localized in different cellular compartments and perform various functions. The onset of cell elongation is coupled with an up-regulation of genes for a subset of lectins, which is strongly enriched in proteins with kinase domains and predicted to be localized in the plasma membrane. Lectin domains of such kinases have direct access to cell wall glycans and can modulate and sense their inevitable rearrangements during cell surface expansion.

An analysis of the expression of genes encoding proteins with lectin domains in the growing maize root revealed some genes whose high and regulated expression during root growth was expected. For example, these include the maize homologs of FERONIA and HERCULES, which are involved in maintaining the integrity of cell walls, and calreticulins, essential for the intensive synthesis of glycosylated proteins that naturally accompany cell growth. Some families of lectins, such as amaranthins and the legume family, showed neither significant levels of mRNA nor a substantial regulation of it. Among the lectins whose expression was surprisingly high and differentially regulated in growing maize root were EULs, dirigent–jacalins, malectins, GNA and Nictaba. The functions of these proteins and their close homologs have either not been established or have not yet been linked to growth processes. Thus, a comprehensive transcriptome analysis of the expression of all lectin genes allowed us to identify a set of molecular players with a high potential to play important roles in the early stages of root morphogenesis. Further studies on the specificity of individual lectin domains in plants, especially toward cell wall glycans and those metabolic and signaling pathways in which the corresponding proteins are involved, are necessary for progress to be made in plant developmental biology.

## Figures and Tables

**Figure 1 plants-11-01799-f001:**
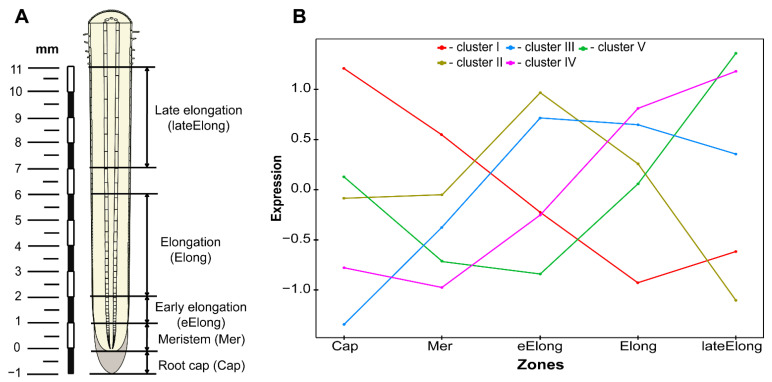
Schematic representation of the maize root zones under study (**A**) and the averaged gene expression profile of the different clusters by zone (**B**).

**Figure 2 plants-11-01799-f002:**
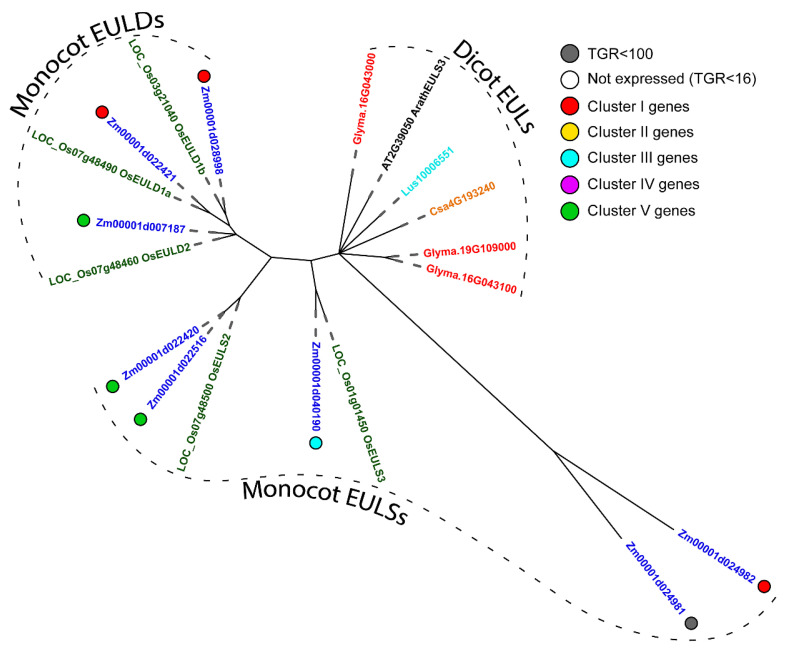
Phylogenetic dendrogram of *Euonymus europaeus* lectin (PF14200) family members of *A. thaliana* (gene names are given in black font), *Z. mays* (blue font), *G. max* (red font), *O. sativa* (green font), *L. usitatissimum* (cyan font) and *C. sativum* (orange font). The colored dots next to *Z. mays* genes denote the character of expression: white—genes with no expression (TGR value less than 16 in all samples); gray—genes with a TGR value less than 100 in all samples; red—genes from cluster I, expression decreased from meristem and root cap zones to late elongation; yellow—genes from cluster II, expression peaks in the early elongation zone; blue—genes from cluster III, high expression from early to late elongation; purple—genes from cluster IV, expression increased in the root cap and also increased from meristem zone to late elongation; green—genes from cluster V, highest expression in the late elongation zone. Only branches with ultrafast bootstrap support values greater than 90 are shown.

**Figure 3 plants-11-01799-f003:**
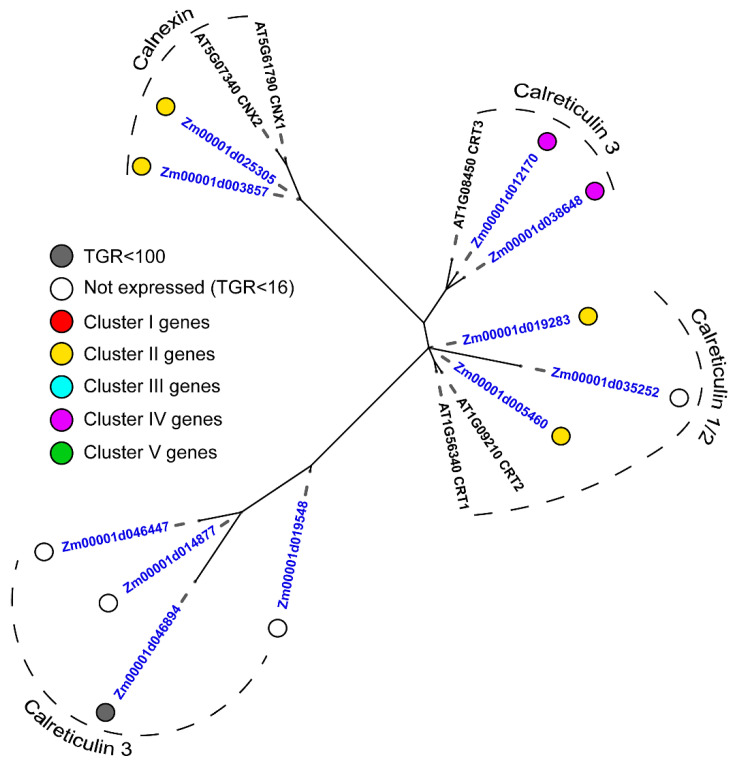
The phylogenetic dendrogram of calreticulin (PF00262) family members of *A. thaliana* (gene names are given in black font) and *Z. mays* (blue font). The colored dots next to *Z. mays* genes denote the character of expression: white—genes with no expression (TGR value less than 16 in all samples); gray—genes with a TGR value less than 100 in all samples; red—genes from cluster I, expression decreased from meristem and root cap zones to late elongation; yellow—genes from cluster II, expression peaks in the early elongation zone; blue—genes from cluster III, high expression from early to late elongation; purple—genes from cluster IV, expression increased in the root cap and also increased from meristem zone to late elongation; green—genes from cluster V, highest expression in the late elongation zone. Only branches with ultrafast bootstrap support values greater than 90 are shown.

**Figure 4 plants-11-01799-f004:**
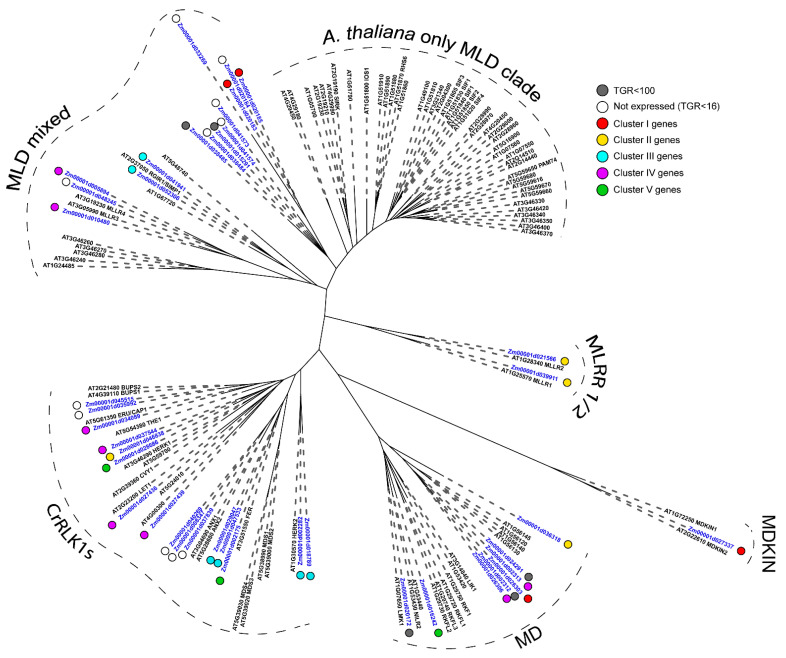
Phylogenetic dendrogram of malectin (PF11721) and malectin-like (PF12819) family members of *A. thaliana* (gene names are given in black font) and *Z. mays* (blue font). The colored dots next to *Z. mays* genes denote the character of expression: white—genes with no expression (TGR value less than 16 in all samples); gray—genes with a TGR value less than 100 in all samples; red—genes from cluster I, expression decreased from meristem and root cap zones to late elongation; yellow—genes from cluster II, expression peaks in the early elongation zone; blue—genes from cluster III, high expression from early to late elongation; purple—genes from cluster IV, expression increased in the root cap and also increased from meristem zone to late elongation; green—genes from cluster V, highest expression in the late elongation zone. Only branches with ultrafast bootstrap support values greater than 90 are shown.

**Figure 5 plants-11-01799-f005:**
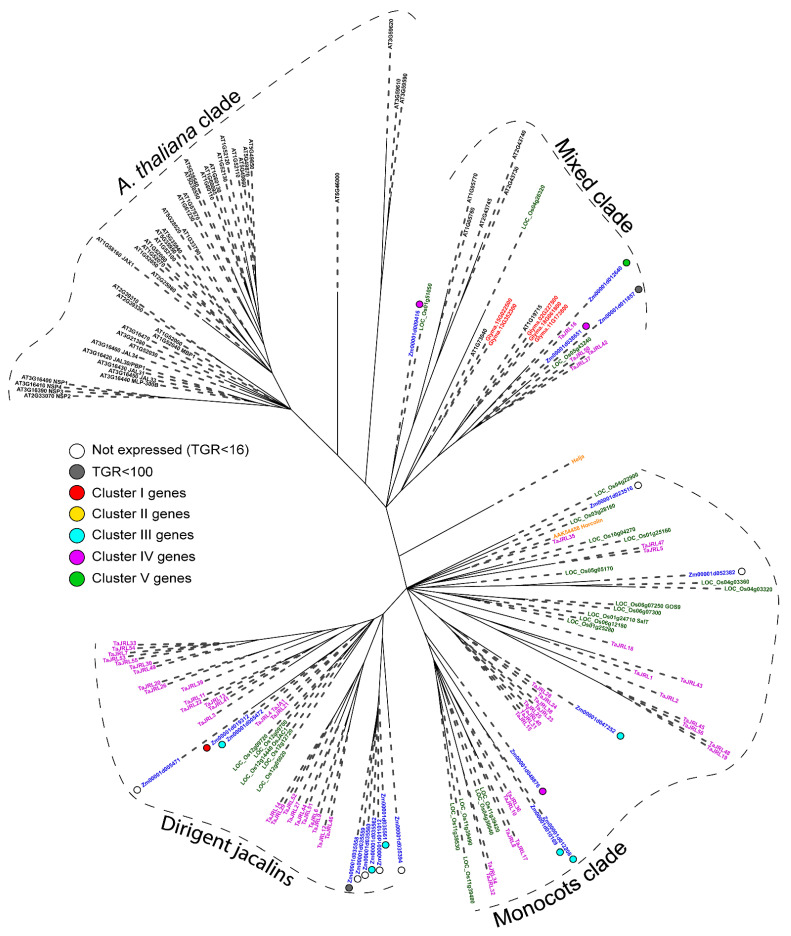
Phylogenetic dendrogram of jacalin-related (PF01419) lectin family members of *A. thaliana* (gene names are given in black font), *Z. mays* (blue font), *G. max* (red font) and *T. aestivum* (purple font). Sequences of two characterized jacalin-related lectins were added for comparison: Helja (from *Helianthus annuus*) and Horcolin (from *Hordeum vulgare*) (UniProtKB P82953) [32]. The colored dots next to *Z. mays* genes denote the character of expression: white—genes with no expression (TGR value less than 16 in all samples); gray—genes with a TGR value less than 100 in all samples; red—genes from cluster I, expression decreased from meristem and root cap zones to late elongation; yellow—genes from cluster II, expression peaks in the early elongation zone; blue—genes from cluster III, high expression from early to late elongation; purple—genes from cluster IV, expression increased in the root cap and also increased from meristem zone to late elongation; green—genes from cluster V, highest expression in the late elongation zone. Only branches with ultrafast bootstrap support values greater than 80 are shown.

**Table 1 plants-11-01799-t001:** Maize lectin families and their expression in the primary root.

Lectin Families	ID (Pfam/cd/ PTHR) ^1^	Number of Genes	Expressed (TGR > 16) ^2^	Kinase Domain	Number of Expressed Genes in Each Cluster (Cutoff 100 TGR/16 TGR)				
					I	II	III	IV	V
GNA	PF01453	81	56	68	5/13	4/7	6/7	13/13	6/16
Legume	PF00139	51	36	46	3/6	2/2	3/3	7/13	6/12
Malectin-like	PF12819	32	21	22	2/2	3/3	6/6	6/6	2/4
LysM	PF01476	29	22	10	3/3	2/2	4/4	8/8	4/5
Jacalin-related	PF01419	20	13	3	1/2	-	6/7	3/3	1/1
Nictaba	PF14299	18	18	0	-	1/2	5/7	6/8	2/2
Galectin-like	PF00337	17	12	0	2/2	4/4	1/1	2/2	1/3
Galactose-binding	PF02140	13	10	0	1/2	3/3	1/1	1/1	2/3
Calreticulin	PF00262	11	7	0	-	4/5	-	2/2	-
Hevein	PF00187	10	7	0	2/2	1/2	-	-	3/3
Malectin	PF11721	9	9	6	2/3	1/2	-	2/2	1/2
EUL	PF14200, PTHR31257	8	8	0	3/4	-	1/1	-	3/3
Amaranthin	PF07468	4	0	0	-	-	-	-	-
RicinB	PF00652	3	3	0	1/1	-	-	1/1	1/1
C-type	PF00059	1	1	1	1/1	-	-	-	-
ABA	PF07367	0	0	0	-	-	-	-	-
CRA	cd02879	0	0	0	-	-	-	-	-
CV-N	PF08881	0	0	0	-	-	-	-	-
Total		307	223	156	26/41	25/32	33/36	51/59	32/55

^1^ Domain IDs are given according to the Pfam (Protein family) and CD (conserved domains) databases and the PANTHER Classification System. ^2^ The number of expressed genes in each gene cluster shown in Figure 1 is given with 100 and 16 TGR (total gene reads) cutoffs.

**Table 2 plants-11-01799-t002:** Top 10 maize genes encoding proteins with lectin domains for each of the gene expression clusters found in growing maize root.

Cluster	Gene ID	Family	Cap	Mer	eElong	Elong	lateElong	Homolog	Name
I	*Zm00001d019312*	Jacalin	24,787	26,384	11,923	2309	5341	*TaJRL4*, *LOC_Os12g14440*	*TaJA1*, *OsJAC1*
I	*Zm00001d028998*	EUL	14,636	7917	2056	2581	6146	*LOC_Os03g21040*	*OsEULD1b*
I	*Zm00001d022421*	EUL	13,268	15,200	4837	5271	15,761	*LOC_Os07g48490*	*OsEULD1a*
I	*Zm00001d026303*	Malectin	3199	1841	830	343	378	*At1g56120*, *At1g56130…*	
I	*Zm00001d021447*	LysM	2833	1071	101	20	7	*At2g17120*	*LYM2*
I	*Zm00001d027803*	Gal-binding	2718	2324	1955	1144	947	*At4g36360*, *At1g45130*	*AtBGAL3/5*
I	*Zm00001d027337*	Malectin	1724	1626	433	60	150	*At2g22610*	*MDKIN2*
I	*Zm00001d024982*	EUL	1648	704	178	21	1	*At2g39050*	*ArathEULS3*
I	*Zm00001d040724*	Galectin-like	1241	1194	1022	666	547	*At3g06440*	*GALT3*
I	*Zm00001d036370*	Hevein	1216	706	447	174	178	*At3g12500*	*CHI-B*
II	*Zm00001d019283*	Calreticulin	19,305	30,606	41,660	21,782	6699	*At1g56340*, *At1g09210*	*CRT1/2*
II	*Zm00001d003857*	Calreticulin	12,625	20,599	30,997	19,097	5488	*At5g61790*, *At5g07340*	*CNX1*, *CNX2*
II	*Zm00001d025305*	Calreticulin	9543	17,954	25,639	10,488	3021	*At5g61790*, *At5g07340*	*CNX1*, *CNX2*
II	*Zm00001d005460*	Calreticulin	3190	6521	8847	3301	944	*At1g56340*, *At1g09210*	*CRT1/2*
II	*Zm00001d046838*	Malectin-like	1217	1881	5314	9648	1813	*At3g46290*, *At2g39360*	*HERK1*, *CVY1*
II	*Zm00001d003190*	Hevein	3935	3508	4029	3823	1784	*At3g54420*	*EP3*
II	*Zm00001d044290*	Gal-binding	1334	1890	2760	1566	738	*At3g13750*	*AtBGAL1*
II	*Zm00001d048440*	Gal-binding	1695	2283	2677	857	496	*At4g36360*, *At1g45130*	*AtBGAL3/5*
II	*Zm00001d049741*	Legume	372	540	2019	2006	176		
II	*Zm00001d030759*	Legume	1311	751	1416	1657	952	*At3g55550*	*LECRK-S.4*
III	*Zm00001d040190*	EUL	1653	4018	10,050	13,790	12,100	*LOC_Os01g01450*	*OsEULS3*
III	*Zm00001d028474*	Gal-binding	905	3794	6665	2806	4568	*At2g28470*	*AtBGAL8*
III	*Zm00001d029047*	Malectin-like	1084	1740	6279	11,398	5394	*At3g51550*	*FER*
III	*Zm00001d047533*	Malectin-like	674	959	3336	5279	2055	*At3g51550*	*FER*
III	*Zm00001d043252*	GNA	63	437	3282	6739	4529		
III	*Zm00001d020691*	LysM	131	841	2942	2065	386	*At1g21880*, *At1g77630*	*LYM1*, *LYM3*
III	*Zm00001d018789*	Malectin-like	628	1183	2500	2943	1197	*At1g30570*	*HERK2*
III	*Zm00001d029673*	Nictaba	694	888	1929	3115	1876	*LOC_Os10g37830*	
III	*Zm00001d010169*	Jacalin	2	264	1667	2354	208		
III	*Zm00001d052306*	Malectin-like	1052	1127	1621	1366	1248	*At2g37050*	*RGIR1/SIMP1*
IV	*Zm00001d041880*	Gal-binding	472	628	932	2616	8397	*At2g32810*	*AtBGAL9*
IV	*Zm00001d038648*	Calreticulin	1986	1912	3037	4522	7938	*At1g08450*	*CRT3*
IV	*Zm00001d002313*	Malectin	1197	1244	3005	9336	6326	*At1g56120*, *At1g56130…*	
IV	*Zm00001d021729*	GNA	834	900	2303	5231	4371	*At4g21390*, *At1g91910*	*B120*, *-*
IV	*Zm00001d007789*	GNA	148	148	413	1077	4219		
IV	*Zm00001d021224*	Nictaba	848	666	1430	3516	3846	*At3g53000*	*PP2-A15*
IV	*Zm00001d012170*	Calreticulin	1636	1723	2684	3413	3109	*At1g08450*	*CRT3*
IV	*Zm00001d026306*	Malectin	540	378	1043	4421	3015	*At1g56120*, *At1g56130…*	
IV	*Zm00001d018452*	Nictaba	889	954	1494	2257	2650	*LOC_Os02g56840*	
IV	*Zm00001d027439*	Malectin-like	336	431	1310	3297	2614	*At4g00300*	
V	*Zm00001d042654*	Gal-binding	1275	1364	1319	1977	3103	*At5g63810*	*AtBGAL10*
V	*Zm00001d009936*	Hevein	447	290	233	620	1533	*At3g12500*	*CHI-B*
V	*Zm00001d010971*	Legume	486	400	602	718	1413	*At5g03140*, *At3g53380*	*LECRK-VIII.2/VIII.1*
V	*Zm00001d007187*	EUL	385	160	68	220	1124	*LOC_Os07g48460*	*OsEULD2*
V	*Zm00001d025921*	Legume	527	411	404	640	867	*-*	
V	*Zm00001d036366*	Hevein	1089	65	150	198	853	*At3g12500*	*CHI-B*
V	*Zm00001d053695*	LysM	160	86	85	247	744	*At2g33580*	*LYK5*
V	*Zm00001d017152*	Hevein	682	227	150	468	658	*At3g54420*	*EP3*
V	*Zm00001d038651*	Jacalin	212	141	112	297	537	*LOC_Os05g43240*	
V	*Zm00001d022420*	EUL	21	22	14	67	520	*LOC_Os07g48500*	*OsEULS2*

The clusters are presented in Figure 1. Each top was formed in the basis of the TGR values in the key zone of an individual cluster. TGR values are presented as a heat map, where the minimum is indicated by blue and the maximum by red. Color coding was applied separately for each cluster. The IDs and names of the homolog genes in *A. thaliana* (At), wheat (Ta) and rice (LOC_Os) are given based on the phylogenetic analysis. Blank cells denote genes for which no close homologs were found.

## Data Availability

The data presented in this study are available within the article.

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
