# Peer review of "Growing Maize Root: Lectins Involved in Consecutive Stages of Cell Development"

_plants, 2022, doi:10.3390/plants11141799_

Round 1

Reviewer 1 Report

Article accepted in the current form.

The authors are advised to improve the manuscript in terms of adequate language levels as well as research paper structure. The authors should elaborate more on their findings and discussion compared to other studies. Many sentences do not have the correct punctuation and it is difficult to read the text. All figures are needed for resolution enhancement. Please check the References in-text and end-list for uniformity in style.The conclusion you have provided is quite brief and provides sufficient feedback on the main objectives of your study.

Reviewer 2 Report

I have gone through the research manuscript entitled: “Growing Maize Root: Lectins Involved in Consecutive Stages of Cell Development”. This is a carefully done study and the findings are of considerable interest, originality and thus merit for publication to the Plants. The language of the manuscript is written well and expressed clearly. Especially, describing the relation between specific lectin gene expression in portion of maize roots and the cell development will contribute on plant glycobiology field.

I mentioned my opinion as shown as below:

1)    In Figure1., there are indication of averaged gene expression profile among maize root zones. However, there is no explanation of Cluster I to V. Even there are explanation of ingredients of Cluster at table 2, it is better to mention briefly that what is Clusters (I to V) at Figure 1. This will be helpful to understand the readers.

2)    Is there a Figure 8? P16 line559

3)    There is a sentence, “The abundance of calreticulins 1/2”(P16, line567). In general, calreticulins are known to involve in the quality control of the folding of glycoproteins. It will be helpful to describe the mechanisms of active growth which involve the calreticulins such as explaining what kind of protein are synthesis for instance.

4)    This manuscript has only described about the lectin gene expression among the maize roots. Maize has many protion such as stalks, seeds and leaves. Especially, seeds are very important because it will be connected to agriculture and farming. Finding of new phenomena will contribute to these fields. If the author had added the information of lectin gene expression, it could have overhead view of function of lectins on plant glycobiology. In addition, author should explain the reason why you choose the maize roots for current study.

Reviewer 3 Report

HERK1 has been described as "Hercules receptor kinase 1" not all capitals. 

https://www.ncbi.nlm.nih.gov/gene?Db=gene&Cmd=DetailsSearch&Term=823774

The text nominated that the C-type lectin family is a classical plant lectin (L92). However, commonly this family categorizes as an animal lectin family and it has discrepancies with the description of L122, though a few are reported in Chlamydomonas (Front Plant Sci 10:36, Int J Mol Sci. 18:1164 ). To avoid confusion for many common readers, it would be better to move the category of the C-type lectin family to L97 and put references that the family presents in plants.

The term "Ricin B-like lectin(s)" (https://www.ebi.ac.uk/interpro/entry/InterPro/IPR035992/) sometimes drifts in the text such as RicinB (L95), so they should unity as Ricin B-like lectin or Ricin B-like in the text, tables, and figures.

Defining lectin families in the text, adding the Interpro ID such as Galactose-binding lectin domain (IPR000922) is useful.

"Lectin families" instead of "Family" in the first column will be clear in Table 1.

PTHR31257 in EUL could not reach the link. Because EUL is a member of Ricin B-like lectin, if only PF14200 is enough, PTH.. should delete.

"cd02879" in CRA (https://www.uniprot.org/uniprotkb/B7FNG8/entry#family_and_domains) could not reach the links. Please update the correct one (http://pfam.xfam.org/family/PF00704)

EUL, Amarantin, and Ricin B-like lectin are common β-trefoil fold ing lectin that belongs to the R-type lectin family (https://www.ncbi.nlm.nih.gov/books/NBK579951/).

To help the understand readers, these relationships should write in the text with references.
